# IL-10 Promotes CXCL13 Expression in Macrophages Following Foot-and-Mouth Disease Virus Infection

**DOI:** 10.3390/ijms24076322

**Published:** 2023-03-28

**Authors:** Zijing Guo, Fei Chen, Shuaiyang Zhao, Zhixiong Zhang, Huijun Zhang, Ling Bai, Zhidong Zhang, Yanmin Li

**Affiliations:** 1College of Animal Husbandry and Veterinary Medicine, Southwest Minzu University, Chengdu 610041, China; 2State Key Laboratory on Veterinary Etiological Biology, Lanzhou Veterinary Research Institute, Chinese Academy of Agricultural Sciences, Lanzhou 730030, China

**Keywords:** foot-and-mouth disease virus, C-X-C motif chemokine ligand 13, interleukin-10, macrophage, signaling pathways

## Abstract

Foot-and-mouth disease (FMD) is one of the most contagious livestock diseases in the world, posing a constant global threat to the animal trade and national economies. The chemokine C-X-C motif chemokine ligand 13 (CXCL13), a biomarker for predicting disease progression in some diseases, was recently found to be increased in sera from mice infected with FMD virus (FMDV) and to be associated with the progression and severity of the disease. However, it has not yet been determined which cells are involved in producing CXCL13 and the signaling pathways controlling CXCL13 expression in these cells. In this study, the expression of CXCL13 was found in macrophages and T cells from mice infected with FMDV, and CXCL13 was produced in bone-marrow-derived macrophages (BMDMs) by activating the nuclear factor-kappaB (NF-κB) and JAK/STAT pathways following FMDV infection. Interestingly, CXCL13 concentration was decreased in sera from interleukin-10 knock out (IL-10^-/-^) mice or mice blocked IL-10/IL-10R signaling in vivo after FMDV infection. Furthermore, CXCL13 was also decreased in IL-10^-/-^ BMDMs and BMDMs treated with anti-IL-10R antibody following FMDV infection in vitro. Lastly, it was demonstrated that IL-10 regulated CXCL13 expression via JAK/STAT rather than the NF-κB pathway. In conclusion, the study demonstrated for the first time that macrophages and T cells were the cellular sources of CXCL13 in mice infected with FMDV; CXCL13 was produced in BMDMs via NF-κB and JAK/STAT pathways; and IL-10 promoted CXCL13 expression in BMDMs via the JAK/STAT pathway.

## 1. Introduction

CXCL13 (C-X-C motif chemokine ligand 13) has an important role in the regulation of primary and secondary follicles [1]. CXCL13 signals could induce the development of secondary lymphoid tissue and the trafficking of B cell and follicular T helper (Tfh) cells to germinal centers (GCs) via C-X-C chemokine receptor type 5 (CXCR5) [2]. CXCL13 also contributes to the regeneration of ectopic lymphoid structures in nonlymphoid organs in chronic inflammatory diseases (e.g., chronic obstructive pulmonary disease) and in autoimmune diseases (e.g., rheumatoid arthritis) [3,4,5]. Until now, CXCL13 has been reported to be produced by follicular dendritic cells (FDCs) and stromal cells [6,7], T cells [8,9], B cells [4], human alveolar macrophages (AM) [10], lung fibroblasts [11] and monocyte-derived macrophages (MoDM) [10,12]. In general, FDCs and stromal cells are the main sources of CXCL13 in normal and inflamed lymphoid tissues. However, in ulcerative colitis rheumatoid and arthritis lesions, CXCL13 was produced in macrophages but not in FDCs and stromal cells [10]. Thus, it seems that CXCL13 could be produced by different cells in different diseases.

Recently, the level of CXCL13 in sera has been reported to be elevated in several viral infectious diseases, including severe acute respiratory syndrome coronavirus 2 (SARS-CoV-2) [13], human immunodeficiency virus (HIV) [14], and hepatitis B virus (HBV) infections [15]. It was important to note that both the elevated levels in sera and the increased expression in tissues of CXCL13 were associated with severe prognosis and increased mortality following these viral infections; therefore, CXCL13 has been proposed as a biomarker for predicting disease progression [13,14]. It was reported that CXCL13 was a benefit for virus clearance through forming ectopic GCs, which could generate distinct broadly neutralizing protective antibody responses [11,15]. However, the persistently elevated levels of CXCL13 in the tissue and sera may be detrimental. They could be responsible for promoting the inflammation and facilitating tissue fibrosis, such as lung inflammatory and even lung fibrosis [16]. These effects may be related to the type of cells produced, the levels of available CXCL13, the time point at which CXCL13 increases, etc.

Foot-and-mouth disease virus (FMDV) is a single-stranded positive-sense RNA virus, which is the causative agent of foot-and-mouth disease (FMD). FMD is a highly contagious vesicular disease of livestock, which can cause significant economic losses to the pig and cattle industries [17]. Lymphopenia refers to lymphocyte depletion in the peripheral blood, and it is associated with immune suppression during FMDV infection. Our recent study found that interleukin-10 (IL-10) and CXCL13 were obviously elevated in sera from mice infected with FMDV and were related to lymphopenia and the progression and severity of the disease [18]. It was noted that IL-10 mediated lymphopenia in mice infected with FMDV [18]. However, it is still unclear whether IL-10 regulates CXCL13 during FMDV infection and which cells are involved in producing CXCL13. Therefore, the study aimed to investigate whether IL-10 regulated CXCL13 and to identify the cellular source of CXCL13 during FMDV infection.

This study reported for the first time that macrophages and T cells were the main cellular sources of CXCL13 after FMDV infection. We then identified the signaling pathways that regulated CXCL13 expression in bone-marrow-derived macrophages (BMDMs) after FMDV infection in vitro. Last, we found that IL-10 was an important cytokine which mediated the expression of CXCL13 induced by FMDV infection, and further analysis showed that IL-10 regulated CXCL13 expression in BMDM via the JAK/STAT signaling pathway.

## 2. Results

### 2.1. CXCL13 Was Expressed in Macrophages and T Cells from the Spleen of FMDV-Infected Mice

Our recent study found that the acute infection of FMDV resulted in the death of all mice within 72 h post-inoculation (hpi) [18]; thus, we first analyzed the mRNA expression of CXCL13 in the heart, liver, spleen, lung, and kidney collected from mock mice and FMDV-infected mice at 12, 24, 36, 48, 60, and 72 hpi. The results showed that the levels of the CXCL13 mRNA expressed in the lung, liver, and kidney from infected mice were similar to that in those tissues from mock mice. However, the levels of the CXCL13 mRNA expressed in the spleen and heart of infected mice at 48, 60, and 72 hpi were significantly upregulated compared with those in the heart and spleen of mock mice (Figure 1A). To determine the cellular source of CXCL13 and considering the difficulty in isolating T cells, B cells, natural killer cells (NK cells), macrophages, and dendritic cells (DCs) from the heart, these cells were sorted from the spleen of infected mice at 48 hpi and mock mice with flow sorting (Appendix A), and then the levels of CXCL13 mRNA expression in these cells were quantified by reverse transcription-quantitative polymerase chain reaction (RT-qPCR). The results showed that the levels of CXCL13 mRNA expression in B cells, NK cells, and DCs sorted from the spleen of infected mice were not different from that in those cells sorted from mock mice (Figure 1B). However, the levels of CXCL13 mRNA expression in macrophages (average 4.7-fold, *p* < 0.01) and T cells (average 3.8-fold, *p* < 0.05) from infected mice were significantly upregulated compared with that of macrophages and T cells from mock mice, respectively (Figure 1B). To further determine whether CXCL13 was produced by macrophages and T cells during FMDV infection, CXCL13 was co-stained with F4/80 or CD3, which are considered as the markers for macrophages and T cells, respectively [19]. Multiplex immunofluorescence staining showed that CXCL13 was less present in macrophages and T cells from spleen of mock mice (Figure 1C,D). However, CXCL13 was obviously located in macrophages and T cells from the spleen of infected mice (Figure 1E,F), and the fluorescence intensity of the CXCL13 was significantly increased in these macrophages and T cells from the spleen of infected mice (both *p* < 0.0001) (Appendix A). These results indicated that the elevated CXCL13 was mainly produced by macrophages and T cells from mice following FMDV infection.

### 2.2. CXCL13 Was Expressed in Mice BMDMs following FMDV Infection

Previous studies suggested that lymphocytes including T cells were not infected with FMDV [18,20,21]; therefore, FMDV infection in vitro was carried out in macrophages to verify the cellular source of CXCL13 in FMDV-infected mice. BMDMs are primary macrophages derived from bone marrow cells cultured in the condition of growth factors in vitro. In this study, BMDMs were isolated from healthy mice and then infected with FMDV. In the preliminary experiment, the levels of CXCL13 mRNA expression in FMDV-infected BMDMs were significantly increased as early as 24 h after infection (Appendix A). Thus, the CXCL13 expression in BMDMs was quantified using RT-qPCR and an enzyme-linked immunosorbent assay (ELISA) at 24 hpi. The results suggested that the expression level of CXCL13 mRNA (average 5.8-fold, *p* < 0.05) was significantly upregulated in BMDMs infected with FMDV compared with that of BMDMs without FMDV infection (Mock group) (Figure 2A). The CXCL13 protein concentration was also significantly increased in the cell culture supernatants of FMDV-infected BMDMs compared with that of the Mock group (Mock group vs. FMDV group: 76.9 ± 3.5 pg/mL vs. 1937.0 ± 136.5 pg/mL, *n* = 3, *p* < 0.001) (Figure 2A). The immunofluorescence experiment further demonstrated that there was no CXCL13 protein detectable in BMDMs without FMDV infection, while CXCL13 protein was presented in FMDV-infected BMDMs at 24 hpi (Figure 2B). Accordingly, in the FMDV-infected BMDMs, the FMDV RNA sequence and VP1 protein were detected, and the FMDV RNA load in BMDMs infected with FMDV was 7.6 ± 1.0 × 10^7^ copies of viral RNA/μg total RNA (*n* = 3) (Figure 2C), indicating that BMDMs could be infected with FMDV in vitro. All these results demonstrated that CXCL13 could be produced in BMDMs after FMDV infection.

### 2.3. IL-10 Mediated FMDV-Induced CXCL13 Production in Mice

Our recent study found that the increased trend in CXCL13 in the sera of infected mice was consistent with that of IL-10 in the sera of infected mice [18]. Thus, we analyzed the expression level of CXCL13 in FMDV-infected mice after being treated with anti-IL-10R antibody in vivo. The results showed that the CXCL13 concentration in sera was significantly more decreased in infected mice with IL-10R blocker than that of mice treated by isotype antibody at 48 hpi (IL-10R group vs. isotype group: 753.8 ± 250.9 pg/mL vs. 2056.0 ± 301.7 pg/mL, respectively, *n* = 10, *p* < 0.01) (Figure 3A). Furthermore, CXCL13 in sera was also significantly more decreased in IL-10^-/-^ mice than that in wild-type mice at 48 hpi (FMDV-IL-10^-/-^ group vs. FMDV-Wild type group: 580.4 ± 186.5 pg/mL vs. 1920.0 ± 223.8 pg/mL, respectively, *n* = 10, *p* < 0.001) (Figure 3B). The CXCL13 concentration was significantly more decreased in IL-10^-/-^ mice than that in wild-type mice (Mock-IL-10^-/-^ group vs. Mock group: 112.1 ± 13.4 pg/mL vs. 203.3 ± 8.2 pg/mL, respectively, *n* = 10, *p* < 0.0001) (Figure 3B). In addition, the expressions of CXCL13 mRNA were significantly decreased in the heart and spleen of mice treated with IL-10R blocker when compared with that of isotype-treated mice at 48 hpi (*n* = 10, *p* < 0.0001) (Figure 3C,D). The CXCL13 mRNA expressions were significantly more reduced in the heart and spleen of IL-10^-/-^ mice than that of wild-type mice (*n* = 10, *p* < 0.01 and *p* < 0.001, respectively) (Figure 3E,F). The expressions of CXCL13 mRNA were also significantly more decreased in the heart and spleen of IL-10^-/-^ mice infected with FMDV than that of wild-type mice infected with FMDV at 48 hpi (*n* = 10, *p* < 0.0001) (Figure 3E,F). These results indicated that knocking out IL-10 or blocking IL-10/IL-10R signaling in vivo could suppress CXCL13 production in mice following FMDV infection.

### 2.4. IL-10 Enhanced CXCL13 Expression in FMDV-Infected BMDMs

We next determined if IL-10 mediated CXCL13 expression in BMDM during FMDV infection. As shown in Figure 4A,B, it was demonstrated that both mRNA and protein of IL-10 were significantly increased in BMDMs after FMDV infection, indicating that FMDV infection induced IL-10 production in BMDMs. To explore if IL-10 could regulate CXCL13 expression in BMDMs following FMDV infection, the anti-IL-10R antibody was used to block IL-10/IL-10R signal pathways in BMDMs in vitro. The results suggested that the cell viability of BMDMs was not affected after IL-10R antibody treatment or isotype antibody treatment (Appendix A). It was noted that the CXCL13 protein level in BMDMs treated by the anti-IL-10R antibody was significantly reduced compared with that in BMDMs treated by the isotype antibody following FMDV infection (isotype group vs. IL-10R group: 2025.2 ± 121.4 pg/mL vs. 1366.0 ± 83.7 pg/mL, respectively, *n* = 3, *p* < 0.05), and the CXCL13 protein concentration was significantly increased in BMDMs stimulated with rIL-10 alone (Mock vs. rIL-10 group: 146.5 ± 33.9 pg/mL vs. 469.6 ± 23.1 pg/mL, respectively, *n* = 3, *p* < 0.01) (Figure 4C). Furthermore, in IL-10^-/-^ BMDMs derived from IL-10^-/-^ mice, the CXCL13 protein concentration was significantly decreased compared that in wild-type BMDMs after FMDV infection (FMDV-Wild type group vs. FMDV-IL-10^-/-^ group: 2158.0 ± 257.7 vs. 1175.1 ± 113.7 pg/mL, respectively, *n* = 3, *p* < 0.05) (Figure 4D). These results supported that IL-10 regulated CXCL13 production in BMDMs after FMDV infection.

### 2.5. FMDV Induced CXCL13 Expression in BMDMs by Activating NF-κB and JAK/STAT Signaling Pathways

We next examined signaling pathways that mediated the expression of CXCL13 in BMDMs during FMDV infection. It had been reported that the nuclear factor-kappaB (NF-κB) signaling pathway was involved in CXCL13 expression in stromal cells [6], DCs [22], and macrophages [12]; thus, we determined if NF-κB signaling pathway was involved in the expression of CXCL13 in BMDMs following FMDV infection. The results showed that the late productions of the canonical and noncanonical NF-κB subunits, including IκBa phosphorylation (p-IκBa), P100, P52, and RelB, were significantly upregulated in BMDMs infected with FMDV at 24 hpi (Figure 5A). However, in BMDMs pretreated by a potent NF-κB inhibitor BAY 11-7082 (BAY) [23], the expressions of p-IκBa, P100, P52, and RelB were decreased (Figure 5A). Further analysis showed that there was no significant difference in the expression levels of IκBa in BMDMs, FMDV-infected BMDMs, BAY-treated BMDMs, and BMDMs infected with FMDV and treated by BAY. However, the gray value ratio of p-IκBa/IκBa in BMDMs infected with FMDV and treated with BAY was significantly decreased compared with that of BMDMs infected with FMDV (*p* < 0.001). It was indicated that BAY inhibited the levels of p-IκBa in BMDMs after FMDV infection, but not IκBa (Figure 5B). The cell viability of BMDMs was not affected after BAY treatment (Appendix A). Notably, BAY significantly abrogated CXCL13 release in BMDMs after FMDV infection at 24 hpi (FMDV group vs. FMDV + BAY group: 1925.0 ± 74.7 pg/mL vs. 536.7 ± 200.7 pg/mL, respectively, *n* = 3, *p* < 0.01) (Figure 5C), which suggested that NF-κB signaling pathway was involved in CXCL13 expression in BMDMs after FMDV infection.

In addition, the expressions of the signal transducer and activator of transcription 3 (STAT3), STAT3 phosphorylation (p-STAT3), and tyrosine kinase 2 (TYK2) were upregulated in BMDMs infected with FMDV at 24 hpi (Figure 5D). We then tested if JAK/STAT signaling pathways also mediated CXCL13 synthesis in BMDMs after FMDV infection. Ruxolitinib, a potent JAK1/JAK2 inhibitor [24], deregulated STAT3, p-STAT3, and TYK2 in BMDMs infected with FMDV (Figure 5D). Similarly, the cell viability of BMDMs was also not affected after ruxolitinib treatment (Appendix A). Further analysis showed that the expression of STAT3 in BMDMs infected with FMDV and treated by ruxolitinib was significantly decreased compared with that of BMDMs infected with FMDV *(p* < 0.0001), and the gray value ratio of p-STAT3/STAT3 in BMDMs infected with FMDV and treated by ruxolitinib was also significantly decreased compared with that of BMDMs infected with FMDV (*p* < 0.0001), which indicated that ruxolitinib inhibited the levels of *p*-STAT3 and STAT3 in BMDMs after FMDV infection (Figure 5E). Similarly, ruxolitinib also significantly abrogated CXCL13 release in BMDMs infected with FMDV at 24 hpi (FMDV group vs. FMDV + Ruxo group: 2058.0 ± 206.8 pg/mL vs. 423.3 ± 133.0 pg/mL, respectively, *n* = 3, *p* < 0.01) (Figure 5F). These results demonstrated that FMDV infection induced CXCL13 production in BMDMs by activating NF-κB and JAK/STAT signaling pathways.

### 2.6. IL-10 Induced CXCL13 Expression in BMDMs by Activating JAK/STAT Signaling Pathway following FMDV Infection

To investigate how IL-10 mediates CXCL13 expression, BMDMs were treated with the rIL-10 and BAY or ruxolitinib at the same time. The results showed that CXCL13 protein concentration in BMDMs followed by the rIL-10 induction and treated with BAY was not different from that in BMDMs followed by the rIL-10 induction alone (rIL-10 + BAY group vs. rIL10 group: 486.7 ± 44.3 vs. 493.1 ± 27.1 pg/mL, respectively, *n* = 3, *p* = 0.9083) (Figure 6A). However, the CXCL13 protein concentration in BMDMs followed with the rIL10 induction and treated with ruxolitinib was significantly decreased compared with that of BMDMs followed with the rIL-10 induction alone (rIL-10 + Ruxo group vs. rIL-10 group: 179.2 ± 24.4 vs. 493.1 ± 27.1 pg/mL, respectively, *n* = 3, *p* < 0.001) (Figure 6A). In addition, the amounts of the P52, NF-κB P65 phosphorylation (p-P65), RelB, and P100 did not significantly change in BMDMs treated with anti-IL-10R antibody following FMDV infection, and these proteins also did not modify the amounts in BMDMs stimulated with rIL-10 alone (Figure 6B). In contrast, the amounts of STAT3, p-STAT3, and TYK2 were obviously decreased in BMDMs treated by anti-IL-10R antibody following FMDV infection, and these proteins were increased in BMDMs stimulated with rIL-10 alone (Figure 6B). Furthermore, the amounts of STAT3, p-STAT3, and TYK2 were decreased in FMDV-infected IL-10^-/-^ BMDMs compared with that in FMDV-infected wild-type BMDMs, rather than P100, RelB, p-P65, and P52 (Figure 6C). Further analysis showed that the gray value ratio of p-STAT3/STAT3 was not significantly different whether inhibiting IL-10 or adding rIL-10, which indicated that p-STAT3 was decreased just after the reducing of STAT3 in FMDV-infected BMDMs following blocking IL-10/IL-10R signaling or knocking out IL-10 (Figure 6D,E). These results indicated that IL-10 regulated CXCL13 expression in BMDMs via activating JAK/STAT rather than the NF-κB signaling pathway following FMDV infection.

## 3. Discussion

CXCL13 is responsible for initiating and maintaining the secondary lymphoid tissues by activating the homing of B and Tfh cells [2]. Recent studies reported that CXCL13 was increased in viral infectious diseases and was associated with the progression and severity of diseases, such as HIV [14], SARS-CoV-2 [13], HBV [15], influenza A virus (IAV) [11], etc. In our previous study, the elevated level in the sera of CXCL13 was related to the severity of the disease following FMDV infections, which indicated that CXCL13 plays a role in the disease progression in FMDV-infected mice [18]. It is well known that FMD is characterized by myocarditis in piglets and calves in acute and rapidly progressing cases [25], and the increased CXCL13 expression in the heart and spleen may contribute to GCs forming or may result in hyperinflammation. It is interesting to further study whether the CXCL13 expression is increased in the natural host infected with FMDV and its role in disease progression. In addition, IL-10 mediated lymphopenia in C57BL/6 mice infected with FMDV, and blocking IL-10/IL-10R signaling in vivo or knocking out IL-10 inhibited the trafficking of B cells in peripheral blood to the heart [18]. The present study found that IL-10 was able to regulate CXCL13 production in vivo and in vitro during FMDV infection. Taken together with a recent result that CXCL13 was responsible for driving B cells trafficking to tissues [2], it was speculated that IL-10 may drive the trafficking of B cells from peripheral blood to the heart of FMDV-infected mice via regulating CXCL13 production, which contributed to lymphopenia in mice during FMDV infection. 

It is well known that virus clearance contributes to the recovery of diseases. Studies showed that IL-10 promoted the expression of CD163, which was the main PRRSV receptor, and thus IL-10 contributed to enhancing PRRSV replication in monocytes [26,27]. We found that blocking IL-10/IL-10R signaling or knocking out IL-10 could decrease FMDV RNA loads in vivo and in vitro (Appendix A), which may help improve the survival rate of FMDV-infected mice [18].

FDCs and stromal cells are the main cellular sources of CXCL13 in normal and inflamed lymphoid tissues [6,7]. However, in ulcerative colitis rheumatoid and arthritis lesions, CXCL13 was mainly produced by macrophages instead of FDCs and stromal cells [10]. It seems that CXCL13 could be produced by different cells in different diseases. In this study, CXCL13 was expressed in macrophages and T cells from FMDV-infected mice, and CXCL13 was produced in BMDMs by activating the NF-κB and JAK/STAT pathways following FMDV infection, which was similar to other studies in other cells from mice and humans [6,12,22]. It has been reported that T cells were not infected with FMDV [18,20,21]; therefore, FMDV may induce T cells to express CXCL13 via indirect ways. A previous study showed that CXCL13 was expressed in MAIT cells, which are subsets of the CD4-CD8-T cell population. Compared with PD-1-MAIT cells, the expression of CXCL13 in PD-1 + MAIT cells was significantly increased, suggesting that PD-1 may play a role in the expression of CXCL13 [8]. Our previous study observed that PD-1 was up-regulated expression in CD8+ T cells from FMDV-infected mice [18]. Thus, whether FMDV induces T cells to express CXCL13 through PD-1 is worth further study. The study firstly identified the cell source of CXCL13 during FMDV infection, which in part explained the high concentration of CXCL13 in mice infected with FMDV.

CXCL13 production has been reported to be mediated by type I IFN in lung fibroblasts and human monocytes [11,28] and by IL-10 and tumor necrosis factor-alpha (TNF-α) in MoDM [12]. In addition, CXCL13 production may be mediated by IL-1 in mice infected with IAV [29], by IL-22 in a mouse model of Sjögren’s syndrome [30] and by TNF-α in the fat [31]. Our recent study found that IL-10 concentration in sera from mice infected with FMDV was significantly increased and coincided with the level of CXCL13 [18]. It was speculated that IL-10 may regulate CXCL13 expression during FMDV infection. This study first reported that IL-10 regulated CXCL13 expression in C57BL/6 mice (in vivo) and in BMDMs (in vitro) after FMDV infection. 

It has been reported that STAT3 is an essential molecule regulating the downstream transcription of target genes induced through IL-10, and IL-10 regulates downstream signaling by activating STAT3 [32,33]. In this study, the expressions of STAT3 and p-STAT3 were decreased in IL-10^-/-^ BMDMs or in BMDMs treated by anti-IL-10R antibody following FMDV infection, and these proteins were significantly increased in BMDMs stimulated with rIL-10 alone; and there was no significant difference in the gray value ratio of p-STAT3/STAT3 whether inhibiting IL-10 or adding rIL-10, suggesting that p-STAT3 was decreased due to reducing of STAT3 in BMDMs following blocking IL-10/IL-10R signaling or knocking out IL-10. It was speculated that IL-10 regulated CXCL13 expression in BMDMs via activating STAT3 after FMDV infection, which was similar to other studies in human MoDM and AM after lipopolysaccharides (LPS) stimulation [12]. 

Considering that CXCL13 was produced in BMDMs through the NF-κB and JAK/STAT pathways, IL-10 mediated CXCL13 production only via the JAK/STAT pathway. Therefore, we investigated the other potential cytokines that may be involved in CXCL13 production in BMDMs via the NF-κB pathway [11,12,28]. The results showed that the levels of IL-6 and IFN-α in BMDMs infected with FMDV were similar to that of mock BMDMs. However, the level of TNF-α was significantly increased in BMFM after FMDV infection compared with that of mock BMDMs (Appendix A). TNF-α, as a major factor, has been reported to control CXCL13 expression in MoDM via the NF-κB pathway [12]. Thus, TNF-α may be induced CXCL13 production in BMDMs after FMDV infection. Differently, our recent study found that TNF-α concentration in sera from mice infected with FMDV was similar to that of mock mice. Taken together, the study of BMDMs (in vitro) and mice (in vivo) have demonstrated that IL-10 played an important role in inducing CXCL13 production during FMDV infection. 

## 4. Materials and Methods

### 4.1. Mice, Cells and Virus

C57BL/6 mice aged 4 to 5 weeks were obtained from the Animal Center for Lanzhou Veterinary Research Institute (LVRI), Chinese Academy of Agricultural Sciences (CAAS). IL-10^-/-^ C57BL/6 mice were bought from Cyagen Biosciences Inc., (Guangzhou, China). BMDMs and IL-10^-/-^ BMDMs were isolated from healthy C57BL/6 mice and IL-10^-/-^ C57BL/6 mice, respectively [34]. Briefly, the monocytes from bone marrow were cultured for 5 days in Dulbecco’s modified Eagle’s medium (DMEM; Gibco, C14190500BT, NY, USA) containing 10% fetal bovine sera (FBS; Gibco, 10270-106) and 100 U/mL penicillin and 50 μg/mL streptomycin (Gibco, 15140-122) and 10 ng/mL granulocyte-macrophage colony stimulating factor (GM-CSF) (Peprotech, AF-315-03, NJ, USA). The FMDV O/BY/CHA/2010 strain was obtained from LVRI, CAAS, and all infectious experiments were carried out in the biosafety level 3 biocontainment laboratory at LVRI, CAAS. 

### 4.2. Antibodies and Chemicals

Anti-phospho-IκBa (2859T), anti-p100/p52 (4882T), anti-RelB (4922T), anti-phospho-STAT3 (9145T), anti-phospho-NF-κB p65 (3033T), anti-Jak1 (50996S), anti-Tyk2 (35615S), anti-STAT3 (9139T), and anti-GAPDH (5174S) primary antibodies were purchased from Cell Signal Technology (CST, Beverly, MA, USA). FITC anti-mouse CD3 (100204), PE anti-mouse CD19 (152408), PE anti-mouse NK1.1 (156504), PerCP anti-mouse CD11c (117326), APC/Cyanine7 anti-mouse CD45 (157204), PE/Cyanine7 anti-mouse/human CD11b (101216), PE anti-mouse F4/80 (123110), Brilliant Violet 421™ anti-mouse F4/80 (123137), and FITC anti-mouse I-A/I-E (107606) were purchased from Biolegend (San Diego, CA, USA). CXCL13 polyclonal antibody (PA5-47018), CD3 monoclonal antibody (17-0032-82), and DAPI (62247) were purchased from Thermo Fisher (MA, USA). Alexa Fluor^®^ 647 F4/80 monoclonal antibody (ab237332), Donkey Anti-Sheep IgG H&L (Alexa Fluor^®^ 488) (ab150177), Goat Anti-Rat IgG H&L (Alexa Fluor^®^ 594) (ab150160), Goat Anti-Rabbit IgG H&L (HRP) (ab205718), Goat Anti-Pig IgG H&L (Texas Red^®^) (ab6775), and Goat Anti-Mouse IgG H&L (HRP) (ab6789) were bought from Abcam (Cambridge, UK). Polyclonal pig anti-FMDV VP1 antibody was obtained from the OIE Reference Laboratory of LVRI. GM-CSF and rIL-10 were bought from PeproTech (Neuilly sur-Seine, France). Ruxolitinib (INCB018424) and BAY-117082 (BAY 11-7821) were purchased from Selleckchem (Houston, TX, USA).

### 4.3. Infection and Preparation of Samples

BMDMs were infected with FMDV at 2 MOI and incubated at 37 °C for 1 h. Then, the supernatant was discarded and washed with phosphate-buffered solution (PBS), and finally, DMEM containing 2% FBS was added to continue the culture. The supernatant of BHK-21 cell culture lysate was used to treat the BMDMs as uninfected controls (Mock). The supernatants of the BMDMs were collected to elevate the levels of cytokines by ELISA, and cells were used to detect the levels of mRNA and protein. Mice were infected with FMDV subcutaneously (0.05 mL) with 5 × 10^5.5^ TCID_50_ per mouse as in the previous study [18]. Mice were inoculated with 0.05 mL PBS as mock mice. At 48 hpi, sera and tissues of the heart, liver, spleen, lung, and kidney were collected and stored at −80 °C until use.

### 4.4. RNA Extraction and cDNA Synthesis

RNA extraction from all tissue samples or BMDMs collected was carried out using TRIzol reagent (TaKaRa Bio Inc., Osaka, Japan). Briefly, 700 μL of TRIzol was added to 1 g of tissue homogenization (MP, FastPrep-24) or 100 μL of sera. The mixture was vortexed and incubated at room temperature for 5 min. Then, the200 μL of chloroform was added and incubated on ice for 5 min. Then, it was centrifuged (12,000× *g*, 4 °C, 15 min). 350 μL of supernatant was transferred into 1.5 mL tubes, and then 350 μL of isopropanol was added into the tubes. The mixture was vortexed and incubated on ice for 10 min, and then it was centrifuged (12,000× *g*, 4 °C, 15 min). Finally, the supernatant was discarded, and the RNAase-free ddH_2_O was used to dissolve the pellet. Nucleic acid quantification was assessed with a spectrophotometer (GE, NanoVue Plus). An mount of 1 μg RNA was transcribed into cDNA using PrimeScript^TM^ RT reagent Kit with gDNA Eraser (Takara) according to the manufacturer’s instructions and stored at −20 °C. 

### 4.5. RT-qPCR

FMDV RNA isolated from the infected BMDMs was analyzed by RT-qPCR, and the primers were as follows: FMDV 3D-forward primer (F): 5′-ACTGGGTTTTAYAAACCTGTGATG-3′; FMDV 3D-reverse primer (R): 5′-TCAACTTCTCCTGKATGGTCCCA-3′; FMDV 3D-Probe: 5′-ATCCTCTCCTTTGCACGC-3′ [35]. The amplification was conducted in a 20 μL reaction volume containing 0.5 μL of forward primer, 0.5 μL of reverse primer, 0.5 μL of probe primer, 1.5 μL of cDNA, 10 μL of Premix Ex Taq™(Probe qPCR) (Takara, RR390Q), and 7 μL of RNAase-free ddH_2_O. The PCR conditions were as follows: 95 °C for 30 s, 40 cycles of 95 °C for 5 s, and 60 °C for 30 s. The mRNA expression levels of CXCL13 and IL-10 were detected with RT-qPCR. The primer sequences were as follows; CXCL13-F: GCCCAGAGGGCAAAACAAG; CXCL13-R: TGCGGATGGGCTCATAGTCT [36]. IL-10 primers were designed by Pick Primers of the National Center for Biotechnology Information (NCBI). The primers were as follows: IL-10-F: GGGGGCGAGTGTAACAAGAC; and IL-10-R: GGTCACACCATTTGCTGGGT. The RT-qPCR data were normalized using the housekeeping gene GAPDH as an internal control [37]. The expression levels of the GAPDH had been analyzed in different experimental conditions to evaluate the validity of the reference genes. GAPDH (Mm01205647_g1) primers were purchased from Life Technologies. The amplification was conducted in a 20 μL reaction volume containing 0.5 μL of forward primer, 0.5 μL of reverse primer, 1.5 μL of cDNA, 10 μL of TB Green Premix ExTaq reagents (Takara, RR820A), and 7.5 μL of RNAase-free ddH_2_O. The PCR conditions were as follows: 95 °C for 30 s, 40 cycles of 95 °C for 5 s, and 60 °C for 30 s. The melting curve was analyzed using the default parameters. The 2^−ΔΔCT^ method was used to calculate the relative mRNA expression [38]. There were three samples within each group at least, and all the experiments were repeated three times. The RT-qPCR experiment was conducted according to MIQE principles [39].

### 4.6. Cell Sorting and RNA Analysis

The splenic lymphocytes of mice were isolated using Mouse Spleen Lymphocyte Isolation Kit (Tianjin, China). The splenic lymphocytes were stained and then sorted using a BD FACSAriall (BD Biosciences, NJ, USA). Cell subsets were defined as follows: T cells (CD3+), B cells (CD19+), NK cells (NK1.1+), dendritic cells (DCs; CD45+, MHC-Ⅱ+, F4/80−, CD11c+) and macrophages (F4/80+ and CD11b+). Using a BD FACSAria II (BD Biosciences, USA) 5 × 10^5^ cells were sorted. These sorted cells were lysed in lysis buffer provided by a Power up SYBR CELLS-TO-CT kit (Invitrogen, A35381, CA, USA) and then used for RNA analysis, according to the manufacturer’s instructions (Invitrogen, A35381).

### 4.7. Immunofluorescence

The spleen and heart of mice were collected and stored in OCT at −80 ℃. The 4 μm sections of tissue were sliced with a microtome (Leica CM1950, Leica Biosystems, Weztla, Germany). These tissue sections were transferred to charged slides (Matsunami PRO, PRO-01) and air-dried for 20 min. BMDMs were cultured in glass-bottom dishes (NEST, 801001). After washing with Ca^2+^Mg^2+^ free PBS three times, these tissue sections or BMDMs were fixed for 15 min with 4% paraformaldehyde (PA, Solarbio, P1110, Beijing, China) and permeabilized for 15 min with 0.1% Triton™ X-100 (Solarbio, T8200). Then, they were incubated with 5% bovine sera albumin (BSA, MP, 9048–46-8) for 1 h at room temperature. Next, these tissue sections or BMDMs were incubated with primary antibodies (anti-CXCL13 and anti-F4/80 or anti-CD3 antibodies, respectively) overnight at 4 °C and then incubated with fluorochrome-conjugated secondary antibodies at room temperature for 1 h. Finally, these tissue sections and BMDMs were nuclear-stained with DAPI. The fluorescence signals were visualized with a TCS SP8 confocal fluorescence microscope (Leica SP8, Mannheim, Germany). The fluorescence intensity was analyzed by Image J software (Version 1.8.0). At least three independent experiments were performed.

### 4.8. Western Blotting

BMDMs were added with RIPA (Beyotime, P0013B, Shanghai, China) containing 1% protease and phosphatase inhibitor (Thermo Fisher Scientific, A32959) for lysis. Then, the lysates were incubated on ice for 15 min and centrifuged (12,000× *g*, 4 °C, 15 min). Protein quantification was performed according to the instructions of the BCA protein assay kit (Thermo Fisher Scientific, 23225). Loading buffers were added into the lysates and heated at 98 °C for 10 min. Then, it was loaded on a 4% stacking gel and separated on a 10% gel with SDS electrophoresis at a low temperature. The proteins were transferred to polyvinylidene fluoride (PVDF) membranes (Millipore, ISEQ00010, MA, USA). These membranes were incubated in 5% BSA for 1 h at room temperature and then incubated with primary antibody at 4 °C overnight. After washing with PBS-Tween 20, these membranes were incubated with HRP-conjugated secondary antibody for 1 h at room temperature. After washing with PBS-Tween 20, these membranes were used with chemiluminescence (Thermo Scientific™, 34095 and 34577) for visualization. Bands were observed by GE healthcare Amersham™ Imager 600 in the automatic exposure model. The relative density was analyzed by Image J software (Version 1.8.0). At least three independent experiments were performed.

### 4.9. In Vivo Antibody Treatment

Mice were injected intraperitoneally (i.p.) with 200 μg of in vivo anti-mouse IL-10R antibody (BioXCell, BE0049) on days −1, and 0 of infection, and mice were injected intraperitoneally (i.p.) with the same dose of the in vivo rat IgG1 isotype antibody (BioXCell, BE0048) as the isotype control, as described in previous studies [18,40]. 

### 4.10. ELISA

CXCL13 (MBS2019951), TNF-α (MBS175787), IL-6 (MBS2708234), and IFN-α (MBS2022040) ELISA kits were bought from MyBioSource (San Diego, CA, USA), and IL-10 ELISA kit (SEA056Mu) was bought from Cloud Clone Corp (Wuhan, China). The levels of these cytokines in sera and the supernatant of BMDMs were detected according to the manufacturer’s instructions.

### 4.11. Statistical Analysis

All statistical analyses were carried out with the GraphPad Prism version 5.03 software. Data were expressed as mean ± standard deviation (SD). The unpaired *t*-test was used to analyze the significance of the differences between different treatment groups. Throughout this article, *, **, ***, and **** indicate statistically significant differences, with *p* values of <0.05, 0.01, 0.001, and 0.0001, respectively, while ns indicates a non-statistically significant difference (*p* > 0.05).

## 5. Conclusions

CXCL13, a biomarker for predicting disease progression, was associated with greater severity in the disease. The study demonstrated that macrophages and T cells were the main cellular sources of CXCL13 in mice infected with FMDV, which explained the high concentration of CXCL13 in mice infected with FMDV. Furthermore, FMDV promoted CXCL13 expression in BMDMs via activating NF-κB and JAK/STAT pathways. It was noted that IL-10 mediated CXCL13 expression in mice and BMDMs after FMDV infection, and IL-10 regulated CXCL13 production in BMDMs via the JAK/STAT pathway. These results not only contribute to the understanding of IL-10-CXCL13 immune regulation but also could provide valuable insight into the immunopathology of FMD.

## Figures and Tables

**Figure 1 ijms-24-06322-f001:**
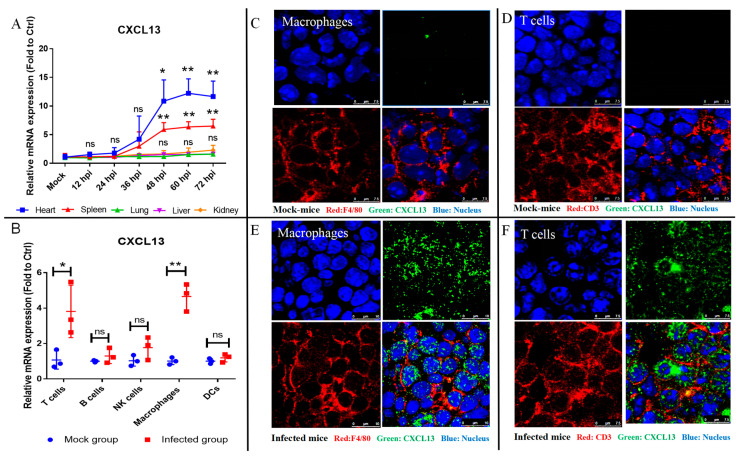
Macrophages and T cells were the cellular sources of CXCL13 in FMDV-infected mice. (**A**) The mRNA expression of CXCL13 in various tissues from mock mice and infected mice at 12, 24, 36, 48, 60, and 72 hpi. (**B**) The mRNA expression of CXCL13 in various cell subsets sorted from mock mice and infected mice. CXCL13, F4/80 (**C**) or CD3 (**D**), and nucleus were co-stained in the spleen from mock mice. CXCL13, F4/80 (**E**) or CD3 (**F**), and nucleus were co-stained in the spleen from infected mice. CXCL13, F4/80, CD3, and nucleus are shown in green, red, red and blue, respectively. At least 10 fields of immunofluorescence were observed. Significance was detected using unpaired *t* test. *, *p* ≤ 0.05; **, *p* ≤ 0.01. While ns indicates a non-statistically significant difference (*p* > 0.05).

**Figure 2 ijms-24-06322-f002:**
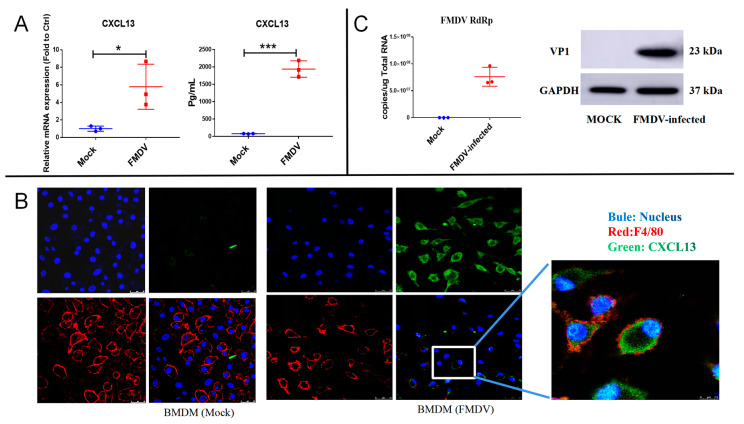
CXCL13 was produced in BMDMs after FMDV infection. (**A**) The mRNA expressions and the protein levels of CXCL13 in uninfected BMDMs (Mock) and FMDV-infected BMDMs at 24 hpi were quantified by RT-qPCR and ELISA, respectively. (**B**) CXCL13 (green), F4/80 (red), and nucleus (blue) were co-stained in Mock and FMDV-infected BMDMs at 24 hpi. (**C**) The FMDV RdRp sequence and FMDV VP1 protein were detected in Mock and FMDV-infected BMDMs at 24 hpi with RT-qPCR and Western blotting, respectively. Upper panels, original magnification ×100 (scale bar = 25 μm). The white box indicated the regions at higher magnifications (scale bar = 7.5 μm). At least 10 fields of immunofluorescence were observed. Significance was detected using unpaired *t* test. *, *p* ≤ 0.05; ***, *p* ≤ 0.001.

**Figure 3 ijms-24-06322-f003:**
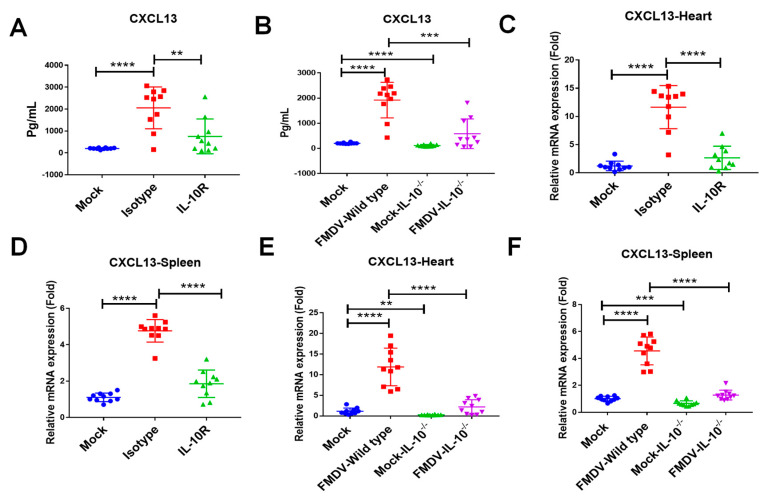
IL-10 mediated CXCL13 expression in FMDV-infected mice. (**A**) CXCL13 concentrations in sera from mock mice, infected mice with anti-IL-10R antibody, and infected mice with isotype antibody were measured by ELISA. (**B**) CXCL13 concentrations in sera from mock mice, wild-type mice infected with FMDV, IL-10^-/-^ mice, and IL-10^-/-^ mice infected with FMDV were measured by ELISA. The mRNA expressions of CXCL13 were detected in the heart (**C**) and spleen (**D**) of mock mice, infected-mice with anti-IL-10R antibody, and infected-mice with isotype antibody at 48 hpi. The mRNA expressions of CXCL13 were detected in the heart (**E**) and spleen (**F**) of mock mice, wild-type mice infected with FMDV, IL-10^-/-^ mice, and IL-10^-/-^ mice infected with FMDV at 48 hpi. Significance was detected using unpaired *t* test. **, *p* ≤ 0.01; ***, *p* ≤ 0.001; ****, *p* ≤ 0.0001.

**Figure 4 ijms-24-06322-f004:**
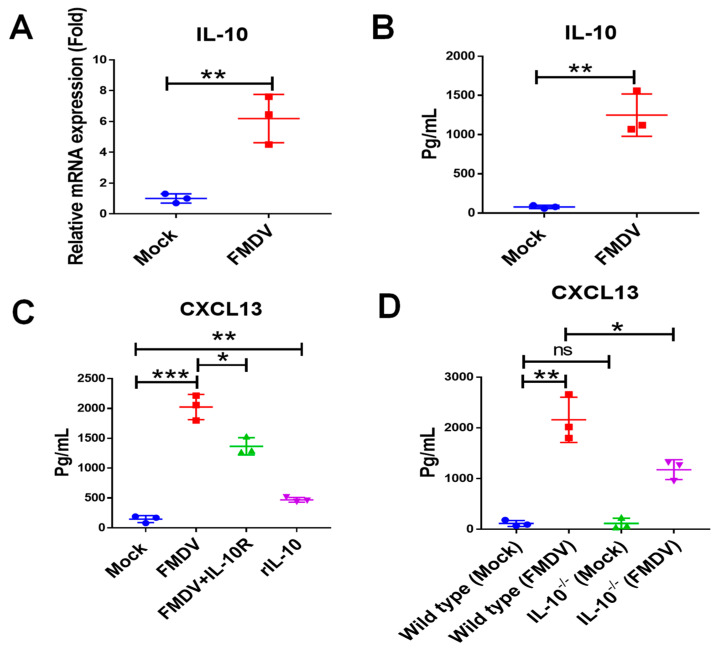
IL-10 controlled CXCL13 expression in BMDMs infected with FMDV. The mRNA (**A**) and protein (**B**) expression of IL-10 in BMDMs infected with FMDV at 24 hpi were measured by RT-qPCR and ELISA, respectively. The CXCL13 concentrations of the supernatant from BMDMs, BMDMs infected with FMDV, BMDMs treated by anti-IL-10R antibody and infected with FMDV, and BMDMs stimulated with rIL-10 at 20 ng/mL (**C**) and from BMDMs, BMDMs infected with FMDV, IL-10^-/-^ BMDMs, and IL-10^-/-^ BMDMs infected with FMDV at 24 hpi (**D**) were measured by ELISA. Significance was detected using unpaired *t* test. *, *p* ≤ 0.05; **, *p* ≤ 0.01; ***, *p* ≤ 0.001. While ns indicates a non-statistically significant difference (*p* > 0.05).

**Figure 5 ijms-24-06322-f005:**
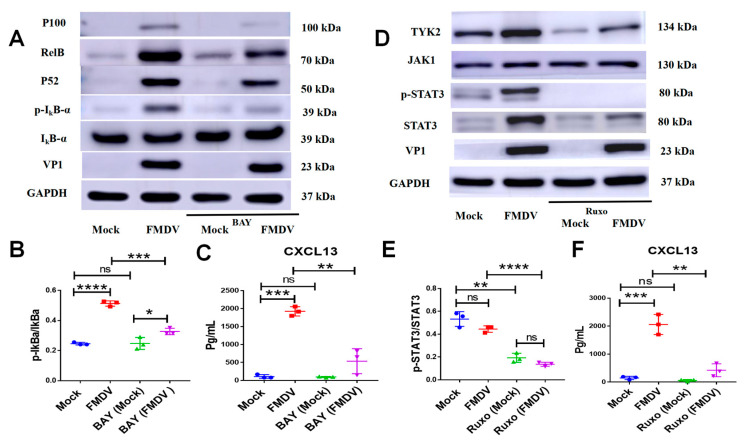
FMDV induces CXCL13 production in BMDMs by activating NF-κB and JAK/STAT signaling pathways. (**A**) The expressions of VP1, IκBa, p-IκBa, P100, P52, RelB and glyceraldehyde-3-phosphate dehydrogenase (GAPDH) in cell lysates of BMDMs, BMDMs infected with FMDV, BMDMs treated by BAY at the recommended concentration of 10 μM, and BMDMs treated by BAY and FMDV were analyzed in by Western blotting. (**B**) The gray value ratio of p-IκBa/IκBa in BMDMs under different conditions of BAY treatment and FMDV infection. (**C**) The CXCL13 concentrations of the supernatant from BMDMs under different conditions of BAY treatment and FMDV infection were measured by ELISA. (**D**) The expressions of VP1, STAT3, p-STAT3, JAK1, TYK2 and GAPDH in cell lysates of BMDMs, BMDMs infected with FMDV, BMDMs treated by ruxolitinib at the recommended concentration of 10 μM, and BMDMs treated by ruxolitinib and FMDV were analyzed by Western blotting. (**E**) The analysis of the gray value ratio of p-STAT3/STAT3 in BMDMs under different conditions of ruxolitinib treatment and FMDV infection. (**F**) The CXCL13 concentrations of the supernatant from BMDMs under different conditions of ruxolitinib treatment and FMDV infection were measured by ELISA. Significance was detected using unpaired *t* test. *, *p* ≤ 0.05; **, *p* ≤ 0.01; ***, *p* ≤ 0.001; ****, *p* ≤ 0.0001. While ns indicates a non-statistically significant difference (*p* > 0.05).

**Figure 6 ijms-24-06322-f006:**
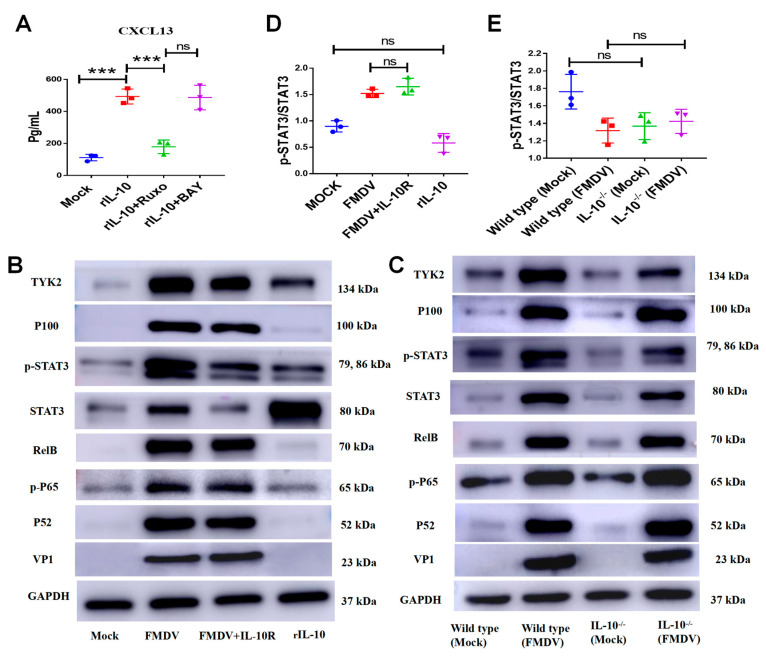
IL-10 induces CXCL13 expression in BMDMs by activating JAK/STAT signaling pathway. (**A**) The CXCL13 concentration in BMDMs followed the rIL-10 induction and treated with BAY or ruxolitinib at the same time. The expressions of VP1, P52, p-P65, RelB, P100, STAT3, p-STAT3, TYK2 and GAPDH in cell lysates of BMDMs, BMDMs infected with FMDV, BMDMs treated by anti-IL-10R antibody and infected with FMDV, and BMDMs stimulated with rIL-10 at 20 ng/mL (**B**) and in cell lysates of BMDMs, BMDMs infected with FMDV, IL-10^-/-^BMDMs, and IL-10^-/-^ BMDMs infected with FMDV (**C**) were analyzed by Western blotting. The gray value ratio of p-STAT3/STAT3 in BMDMs, BMDMs infected with FMDV, BMDMs treated by anti-IL-10R antibody and infected with FMDV, and BMDMs stimulated with rIL-10 at 20 ng/mL (**D**) and in BMDMs, BMDM infected with FMDV, IL-10^-/-^ BMDMs, and IL-10^-/-^ BMDMs infected with FMDV at 24 hpi (**E**). Significance was detected using unpaired *t* test. ***, *p* ≤ 0.001. While ns indicates a non-statistically significant difference (*p* > 0.05).

## Data Availability

Not applicable.

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
