# Peer review of "IL-10 Promotes CXCL13 Expression in Macrophages Following Foot-and-Mouth Disease Virus Infection"

_ijms, 2023, doi:10.3390/ijms24076322_

Round 1
Reviewer 1 Report
Macrophages are significantly involved in the regulation of the immune response to infections. In particular macrophages affect the inflammatory response of the immune system and also play a significant role in recruitment of further effector cells of the immune system such as B and T cells. The interplay between macrophages, infectious agents and other components of the immune system are highly complex and of great significance both for diagnosis and therapy. Macrophages affect the immune response by producing and releasing a vast array of different cytokines depending on the external and internal stimuli.
It is known that CXCL13 cytokine plays a significant role in recruitment of B- and T-cells in various viral infections. Significant release of CXCL13 has been further observed in infections with Foot-and-mouth disease virus (FMDV). However, the origin and the controlling regulatory mechanisms of the observed CXCL13 is not well understood. The authors of the present study aim to address this lack of knowledge by investigating the cellular origins of CXCL13 and regulatory pathways which control CXCL13 release.
Using mouse models and in vitro cell culture models of isolated cells the authors of the study convincingly show that CXCL13 is released by T cells and macrophages. Regarding the regulatory pathways the authors show in detailed and well-performed experiments that IL-10 regulates CXCL13 via JAK/STAT pathway. CXCL13 release due to FMDV infection is further affected by NF-κB pathway. However, this appears to be under the regulation of a different cytokine other than IL-10.
The presented experiments are adequately designed and well performed. The obtained results are clearly and comprehensively presented and mostly satisfy common scientific practice. Hence the study is sound and supports the authors conclusions in their discussion. Nevertheless, when reporting the results of RT-qPCR experiments such as in Figures 1-4 the minimal standards of the MIQE guidelines should be fulfilled. This includes normalization of the results to verified reference genes and reporting normalization procedure as well as the applied reference genes. Thus, the authors should add the relevant information to the method section before acceptance of the manuscript.
The scientific literature in the filed is adequately reflected in the cited references.
Overall the present study is an interesting and relevant addition to the scientific knowledge and elucidates an incompletely understood aspect of inflammatory control and immune response to FMDV infection.
Therefore, acceptance of the present manuscript is recommended after minor revision.
Reviewer 2 Report
This manuscript investigated how IL-10 plays a role in increasing CXCL13 expression in macrophages during foot and mouth disease virus infection. The paper aligns well with the journal's focus and may be publishable after addressing some comments.
Comments and Suggestions:
1) In 2.1, the authors looked at CXCL13 expression at 48 h. did the author find an increase in expression at an early or late time point?
2) In Figures 1 A &B, label the Y axis. Is the fold change normalized to the housekeeping gene?
3) In figures 1 C, D, E & F, Provide the quantification of the figures and also mentions how many fields are observed before making the conclusion. Also, label the type of cells used in these experiments in the figure.
4) In 2.2, why is 24 h used for in vitro experiment? Mention any significance of the timepoint with disease pathogenesis.
5) in vitro and in vivo should be italics?
6) In Figure 2 B, not all BMDMs are stained with F4/80. comment?
7) The English needs improvement; there are typos in the manuscript, for example, on line 151.
8) Treating BMDMs with anti-IL10 R antibody and in using mice lacking IL-10, there is decrease expreson of CXCL13. Did the authors examine the FMDV number, BMDMs viability, and mice survival in these cases?
9) Did BAY and Ruxolitinib treatment impact FMDV number and cell viability?
10) Include more supporting details and deeper understanding in the discussion and conclusion section.
Reviewer 3 Report
Major Comments:
1. In the methods it is mentioned that western blots are developed with chemiluminescence. Though from the original blot images, it does not look like chemiluminescence was used. Kindly clarify. Molecular weight markers are missing from some blots.
2. All graphs in figures should represent the individual values like it is represented in Figure 3.
3. It is mentioned that the expression of CXCL13 was found in both macrophages and T cells from mice infected with FMDV. Authors should discuss the probable relation between CXCL13 expression upregulation in T cells with FMDV though previous studies suggested that lymphocytes including T cells were not infected with FMDV.
Minor Comments:
1. Please mention all unabbreviated forms where it was written first time.
